# PRE-TRAINED LANGUAGE MODELS CAN BE FULLY ZERO-SHOT LEARNERS

## ABSTRACT

How can we extend a pre-trained model to many language understanding tasks, without labeled or additional unlabeled data? Pre-trained language models (PLMs) have been effective for a wide range of NLP tasks. However, existing approaches either require fine-tuning on downstream labeled datasets or manually constructing proper prompts. In this paper, we propose **n**on**p**arametric **prompt**ing PLM (NPPrompt) for fully zero-shot language understanding. Unlike previous methods, NPPrompt uses only pre-trained language models and does not require any labeled data or additional raw corpus for further fine-tuning, nor does it rely on humans to construct a comprehensive set of prompt label words. We evaluate NPPrompt against previous major few-shot and zero-shot learning methods on diverse NLP tasks: including text classification, text entailment, similar text retrieval, and paraphrasing. Experimental results demonstrate that our NPPrompt outperforms the previous best fully zero-shot method by big margins, with absolute gains of 12.8% in accuracy on text classification and 15.6% on the GLUE benchmark. Our source code is available at https://anonymous.4open.science/r/NPPrompt.

## 1 INTRODUCTION

Natural language understanding (NLU) has been important in many applications such as intelligent dialog assistants, online search, and social media analysis. Recent advancement of NLU has been driven by emergent pre-trained language models (PLMs) including BERT (Devlin et al., 2019; Liu et al., 2019b), GPT (Radford et al., 2018; 2019; Brown et al., 2020), BART (Lewis et al., 2020), and T5 (Raffel et al., 2020). Prior studies show that PLMs obtain substantial knowledge during pre-training on raw text corpus (Petroni et al., 2019; Feldman et al., 2019). By fine-tuning on task-specific labeled data, PLMs exploit such knowledge and gain impressive accuracy on a wide range of NLP tasks, such as text classification (Kowsari et al., 2019), question answering (Rajpurkar et al., 2016), machine reading comprehension (Campos et al., 2016), etc.

However, fine-tuning approaches are expensive. It requires labeled datasets, which are rarely available for many tasks. Significant computational efforts are needed to update PLMs' parameters for multiple tasks. In addition, fine-tuning results in one distinct model for each task to maintain.

How can we generalize a pre-trained model to many NLP tasks, without labeled or additional unlabeled data? Existing few-shot and zero-shot approaches propose to construct prompts to elicit desired predictions from PLMs (Brown et al., 2020). The main idea of prompting PLMs is to convert an input utterance to one with masked templates. For example, in text classification an input can be "The Warriors won the NBA championship 2022" and it is instead converted to "A [MASK] news: The Warriors won the NBA championship 2022". A PLM (e.g. BERT) takes the converted text and produces predictions for the masked token, along with the probability. Ideally, a PLM will generate a higher probability for the word "sports" than "politics" on the [MASK] token.

Although these prompting-based methods are effective, they require unlabeled data for training or huge human efforts to construct prompts and to choose designated tokens to represent class labels (Schick & Schütze, 2021a;b; Gao et al., 2021). In addition, these manually constructed *verbalizers*, i.e. mapping from words (e.g. "basketball") to class labels (e.g. SPORTS), do not extend to new emerging categories after PLMs are deployed.

In this paper, we investigate the fully zero-shot learning problem for NLU where only the target label names are available but not the extra raw text. We propose **n**on**p**arametric **prompt**ing PLM (NPPrompt), a novel method to generate predictions for semantic labels without any fine-tuning. NPPrompt uses PLM's own embeddings to automatically find relevant words to labels (e.g. "basketball" and "NBA" for SPORTS), therefore it does not need humans to construct verbalizers. Our key idea is to search for the top $k$ nearest neighbors to a label name in the embedding manifold and then generate and aggregate PLM's predicted logits from masked prompts. In the above case, both predicted values for "basketball" and "NBA" contribute to the final prediction for the SPORTS category. In this way, NPPrompt can be easily generalized to any new categories as long as the category names are semantically meaningful.

The contributions of this paper are as follows. a) We develop NPPrompt, a novel method for fully zero-shot learning with PLMs. b) We conduct extensive experiments on diverse language understanding tasks including text classification, text entailment, similar text retrieval, and paraphrasing. Experimental results show that NPPrompt outperforms the previous zero-shot methods by absolute 12.8% in accuracy on text classification and 15.6% on the GLUE benchmark. Surprisingly, NPPrompt is on a par with the best prior method that trained with manual verbalizers, an additional knowledge base, and extra unlabeled data.

## 2 RELATED WORK

**Prompting** The success of GPT-3 (Brown et al., 2020) has attracted much attention to prompting engineering, a new way to leverage pre-trained language models. Brown et al. (2020) concatenate a few input and output pairs and feed them to the large-scale GPT-3 language model, which is an intuitive in-context learning paradigm, allowing the model to generate answers for additional cases autoregressively. Recent works (Schick & Schütze, 2021a;b) show that small-scale pre-trained language models such as BERT (Devlin et al., 2019), RoBERTa (Liu et al., 2019b) and ALBERT (Lan et al., 2019) can also achieve decent performance using prompt-tuning. Prompting has been applied to a large variety of tasks such as Text Classification (Schick & Schütze, 2021a), Natural Language Understanding (Schick & Schütze, 2021b), Knowledge Probing (Petroni et al., 2019), and Relation Extraction (Han et al., 2021). Typically, a piece of prompt contains a template and a verbalizer. The language model predicts a probability distribution over vocabulary given the template and the verbalizer transforms it into a prediction over class labels. In this work, we focus on designing the verbalizers automatically.

**Verbalizer Design** The verbalizer is an important component in prompting which bridges model outputs and labels and greatly impacts the performance. Schick & Schütze (2021a) design human written verbalizers for prompting, however, they are highly biased towards personal vocabulary with inadequate coverage. Apart from manually designed verbalizers, some recent studies explore automatic verbalizer construction. Auto-L (Gao et al., 2021) uses re-ranking to find the label words set by fine-tuning the model on the candidates searched by RoBERTa; AutoPrompt (Shin et al., 2020) applies gradient-based search to create both prompts and label words automatically with a few trigger examples. But these approaches need to update parameters with gradient descent, which turns out to be infeasible without access to the model weights (e.g., GPT-3). KPT (Han et al., 2021) incorporates external knowledge into the verbalizer in which the unlabeled dataset is needed to refine the label words and thus is not applicable to scenarios where only label names are known. In contrast, our approach NPPrompt directly finds, without any gradient update, relevant words to label names with only PLM's initial word embedding.

**Zero-shot Text Classification** The general zero-shot text classification usually focuses on classifying texts into classes that are unseen during the training process. Transferring knowledge from seen classes to unseen ones requires accurate and discriminative descriptions of all classes (Liu et al., 2019a; Xia et al., 2018), joint embeddings of categories and documents (Nam et al., 2016) or semantic correlations among classes (Rios & Kavuluru, 2018; Zhang et al., 2019). However, these methods require supervised data for the known label set and thus cannot be extended to settings where no labeled pairs for any category is available. (Meng et al., 2020) propose the LOTClass model that uses label names with self-training to do zero-shot classification. But LOTClass still requires unlabeled corpus for extracting the topic-related words and performing self-training. In this work, NPPrompt achieves competitive and even better performance without using any unlabeled dataset.

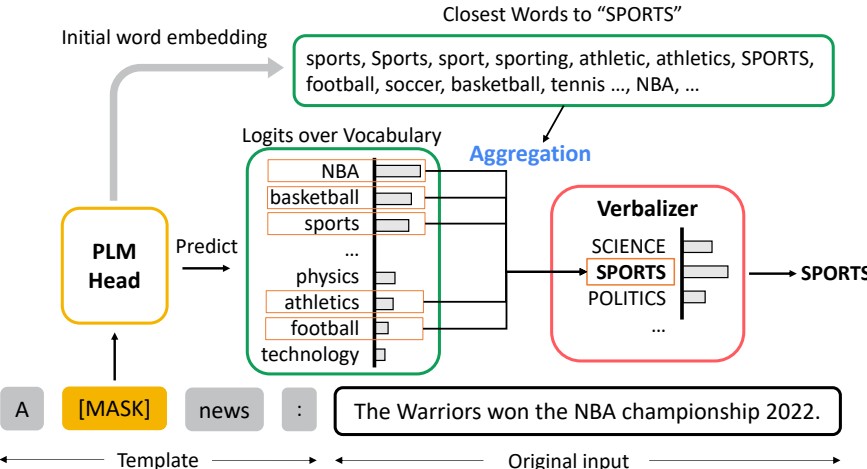

Figure 1: The illustration of NPPrompt. We generate the label words by searching the related words from the initial word embedding of the pre-trained language model. By aggregating logits from the label words, we predict the category with the largest score (SPORTS).

## 3 BACKGROUND: PROMPT-BASED TUNING FOR PLMS

We first provide a standard paradigms, prompt-based tuning, that perform well in few-shot scenarios, before introducing our approach for the zero-shot case. Take $N$ way text classification as an example. We aim to predict the label $y \in \mathcal{Y}$ for each sentence, where $\mathcal{Y}$ is the label set with $N$ distinct classes.

Prompt-based tuning tunes PLM using customized prompts (Brown et al., 2020). The regular prompt-based tuning converts a specific task to a cloze-style mask language modeling problem. For each input example $x$ (single sentence or sentence pair), we first apply a task template $\mathcal{T}$ on it, converting original input $x$ to $x_{\text{prompt}}$. For instance, we concatenate the template "$\mathcal{T}(\cdot) = $ A [MASK] news :" with the original input "The Warriors won the NBA championship 2022" and wrap it into:

$$x_{\text{prompt}} = \mathcal{T}(x) = \text{A [MASK] news} : x \qquad (1)$$

The *verbalizer* $f$ in vanilla prompt engineering map a set of selected words $\mathcal{V}$ from the vocabulary to the original label space $\mathcal{Y}$, i.e., $f : \mathcal{V} \to \mathcal{Y}$. Inversely, we use $\mathcal{M}(y_j)$ to denote the *label words* in $\mathcal{V}$ that are mapped into a specific label $y_j$, $\cup_{y_j \in \mathcal{Y}} \mathcal{M}(y_j) = \mathcal{V}$. Then we calculate the probability of label $y_j$:

$$P(y_j \mid x) = g\left(P([\text{MASK}] = v_i \mid x_{\text{prompt}}) \mid v_i \in \mathcal{M}(y_j)\right), \qquad (2)$$

where $g(\cdot)$ is for aggregating the probability of label words into the probability of the label. Then PLMs can be fine-tuned by minimizing the cross-entropy loss with supervised examples.

## 4 PROPOSED METHOD: NPPROMPT

We inherit PLM with verbalizers framework but keep PLM's parameters frozen (Gao et al., 2021). The key idea of NPPrompt is using PLM's word embeddings to automatically construct verbalizers – mapping from words to labels – in a fully zero-shot way. It does not need any additional raw text corpus for fine-tuning. NPPrompt consists of two steps to compute predictions for any labels in a nonparametric form (Figure 1). 1) We search for all label words closely related to each class $y_j$ in PLM's token embedding manifold. 2) Then we use the PLM to predict values for [MASK], filter them using each class's set of label words, and aggregate the properly weighed outputs to produce the final prediction. In the following, we describe NPPrompt for text classification but it generalizes to other language understanding tasks.

$k$-**Nearest-Neighbor Verbalizer Construction** For each class label (e.g. "SPORTS"), we search over the whole vocabulary $\mathcal{V}$ for the top-$k$ words nearest to the label name in the PLM's embedding

space. Here, the distance between words and label names is measured using the cosine similarity score. Other distance metrics work as well and are examined in Section 5. We denote $k$ as the *neighborhood number*. Assuming the embeddings of word $v_i$ and label name $y_j$ are $\mathbf{emb}(v_i)$ and $\mathbf{emb}(y_j)$ respectively, the label words of the verbalizer for $y_j$ are selected by top-$k$ ranking:

$$\mathcal{M}(y_j) = \underset{v_i \in \mathcal{V}}{\text{Top-}k} \left\{ S(\mathbf{emb}(v_i), \mathbf{emb}(y_j)) \right\}, \tag{3}$$

where $S(\cdot)$ is the cosine similarity function: $S\left(\mathbf{emb}(v_i), \mathbf{emb}(y_j)\right) = \frac{\mathbf{emb}(v_i)}{\|\mathbf{emb}(v_i)\|} \cdot \frac{\mathbf{emb}(y_j)}{\|\mathbf{emb}(y_j)\|}$.

Since the PLM is already pre-trained on raw text corpus, it acquires sensible semantic knowledge and relatedness of words in the vocabulary. We use PLM's embedding to search for label words semantically relevant to given label names. For illustration, we show the found label words of two categories in the AG News dataset (Zhang et al., 2015) and the corresponding similarity scores in Table 1. We also extend our verbalizer to support label names with longer expression in Appendix A.1.

| Word | Sim | Word | Sim |
|------|-----|------|-----|
| " sports" | 1.00 | " business" | 1.00 |
| " Sports" | 0.77 | " Business" | 0.78 |
| " sport" | 0.75 | " businesses" | 0.74 |
| " sporting" | 0.68 | "business" | 0.72 |
| " athletics" | 0.65 | "Business" | 0.67 |
| "sports" | 0.65 | " businessman" | 0.59 |
| "Sports" | 0.65 | " corporate" | 0.58 |
| " Sport" | 0.62 | " company" | 0.56 |
| " athletic" | 0.61 | " enterprise" | 0.55 |
| " athletes" | 0.61 | " businessmen" | 0.55 |

Table 1: The top 10 similar words of the RoBERTa-large model for the AG News dataset categories SPORTS and BUSINESS. Sim: cosine similarity scores.

**Nonparametric Aggregation of Prompted Predictions**  For each input text $x$, we construct a prompt-augmented sequence $x_{\text{prompt}} = \mathcal{T}(x)$ with a [MASK] token. We use the PLM to predict tokens for [MASK]. In contrast to previous prompting methods which directly calculate the probability over the surface labels, we use the nearest label words from above to compute the probability for each output label. Only the words in a label's top-$k$ neighborhood will contribute to the class prediction. The contribution from each label word is non-equal.

To be specific, with $\mathcal{T}(x)$, a PLM produces the logit vector $\Theta_{\text{[MASK]}}$ for all possible words at the [MASK] token. Notice that if the whole vocabulary is $\mathcal{V}$, $\Theta_{\text{[MASK]}} \in \mathbb{R}^{|\mathcal{V}|}$. Then we compute the class probability for a label $y_j$ by aggregating the logits filtered by the verbalizer's label words. We use kernel smoothing to aggregate as follows:

$$Q(y_j|x) = \sum_{v_i \in \mathcal{M}(y_j)} w(v_i, y_j) \cdot \Theta(\text{[MASK]} = v_i | x_{\text{prompt}} = \mathcal{T}(x)) \tag{4}$$

Where the weight between label word $v_i$ and class name $y_j$ are defined as:

$$w(v_i, y_j) = \frac{\exp\left(S(\mathbf{emb}(v_i), \mathbf{emb}(y_j))\right)}{\sum_{v_t \in \mathcal{M}(y_j)} \exp\left(S(\mathbf{emb}(v_t), \mathbf{emb}(y_j))\right)} \tag{5}$$

Finally, the best class prediction is selected from the maximum of all labels:

$$\widetilde{y} = \underset{y_j}{\arg\max} \, Q\left(y_j \mid x\right). \tag{6}$$

Notice since we use kernel smoothing on logits instead of probability, $Q$ is also unnormalized probability.

For example, AG News has two classes $y_1 = \{\text{SCIENCE}\}$, $y_2 = \{\text{SPORTS}\}$. From Table 1, the verbalizer for SPORTS $\mathcal{M}(y_1)$ includes label words "sports", "athletics", etc, and the verbalizer for BUSINESS $\mathcal{M}(y_2)$ includes label words "business", "corporate", etc. Given an input text $x$ "The Warriors won the NBA championship 2022", the prompt-augmented sequence $x_{\text{prompt}}$ will be "A [MASK] news : The Warriors won the NBA championship 2022". The PLM computes logits for every word $\Theta(\text{[MASK]} = v | x_{\text{prompt}})$. NPPrompt computes the unnormalized probabilities for SPORTS and BUSINESS, $Q(\text{SPORTS}|x) = w(\text{"sports"}, \text{SPORTS}) \cdot \Theta(\text{[MASK]} = \text{"sports"}|x_{\text{prompt}}) + w(\text{"athletics"}, \text{SPORTS}) \cdot \Theta(\text{[MASK]} = \text{"athletics"}|x_{\text{prompt}}) + \cdots$, $Q(\text{BUSINESS}|x) = w(\text{"business"}, \text{BUSINESS}) \cdot \Theta(\text{[MASK]} = \text{"business"}|x_{\text{prompt}}) + w(\text{"corporate"}, \text{BUSINESS}) \cdot \Theta(\text{[MASK]} = \text{"corporate"}|x_{\text{prompt}}) + \cdots$. If the aggregated prediction $Q$ for SPORTS is larger than BUSINESS, NPPrompt outputs SPORTS.

There are certain conditions where one class has label names containing little semantic meaning or where several keywords are needed to define a label. For instance, in the DBPedia dataset (Lehmann et al., 2015), one class is related to NATURALPLACE, then we can use the keywords {"river", "lake", "mountain"} to represent this class. In this setting, we pick out the keyword with the maximum score calculated by Equation 4 to represent each label first. Then we choose the label with the largest score. We use $\Phi(y_j)$ to denote all keywords in class $y_j$, and the final prediction is :

$$\widetilde{y} = \arg\max_{y_j} \Big( \arg\max_{y' \in \Phi(y_j)} Q\left(y' \mid x\right) \Big). \tag{7}$$

## 5 EXPERIMENT

We conduct extensive zero-shot learning experiments to demonstrate the effectiveness of our method. We present our implementation details together with the main results and address several research questions pertaining to NPPrompt in this section.

### 5.1 DATASETS, PROMPT TEMPLATES, AND EXPERIMENTAL SETUP

We adopt sentiment classification tasks on two datasets, IMDB (Maas et al., 2011) and Amazon (McAuley & Leskovec, 2013), and topic classification tasks on another two datasets, AG News (Zhang et al., 2015) and DBPedia (Lehmann et al., 2015). All datasets are in the English language. For each task, we directly use the

| Dataset | Classification Type | # Classes | # Test |
|---------|--------------------|-----------|--------|
| AG News | News Topic | 4 | 7,600 |
| DBPedia | Wikipedia Topic | 14 | 70,000 |
| IMDB | Movie Review Sentiment | 2 | 25,000 |
| Amazon | Product Review Sentiment | 2 | 400,000 |

Table 2: Dataset statistics.

test set to assess model performances, without incorporating validation or training sets for post-tuning or cherry-picking hand-crafted prompts. The statistics of each dataset are shown in Table 2.

To concentrate on the verbalizer and reduce the influence of templates, we adopt multiple fixed manual templates following Hu et al. (2022). We report the best template used for the RoBERTa-large model in Table 3.

| Dataset | Template |
|---------|----------|
| AG News | A [MASK] news : $x$ . |
| DBPedia | $x_1$ $x_2$ In this sentence, $x_1$ is a [MASK] . |
| IMDB | $x$ All in all, it was [MASK] . |
| Amazon | $x$ All in all, it was [MASK] . |

Table 3: Prompt templates for NPPrompt.

We implement our experiments based on an open-source toolkit OpenPrompt (Ding et al., 2021), which aims to conduct prompt learning easily. We choose RoBERTa-large (Liu et al., 2019b) as our pre-trained language model. We report the best accuracy of classification results for all experiments using different neighborhood numbers. Since we directly use the pre-trained models for testing, there is no randomness (random seed) in this process. All experiments are conducted on Nvidia A6000 GPUs and more details can be found in Appendix A.1.

### 5.2 BASELINES

We evaluate the following baseline methods.

**Semantic Retrieval**   We utilize sentence embedding models (Reimers & Gurevych, 2019) to obtain the embedding for each sentence and descriptions for each class. Then we calculate the cosine similarity between sentences and label descriptions. We assign the most similar class labels to the sentence. Particularly, we use all-mpnet-base-v2 from Hugging Faceas the sentence embedding model, and the descriptions for each class can be found in Appendix A.1.

**NSP-BERT**   Sun et al. (2021) propose text entailment tasks to replace text classification tasks and then use the Next Sentence Prediction (NSP) head to predict the results. We show the template we use in Appendix A.1.

**ManualVerb**   Manual verbalizers are defined by human experts with domain knowledge and we simply use the label words provided by OpenPrompt (Ding et al., 2021).

**LOTClass**   Meng et al. (2020) employ pre-trained neural language models with unlabeled data for category understanding, i.e., finding words similar to label names. They then introduce a self-training approach to the entire unlabeled corpus to generalize the model.

**GPT-3 with descriptions**   Following Brown et al. (2020), we manually write the descriptions for each class and query GPT-3 where the predicted token serves as the prediction. We show the descriptions in Appendix A.1.

**KPT**   Hu et al. (2022) propose knowledgeable prompt-tuning, which expands the label words space using external knowledge bases (KB). KPT also refines the expanded label words based on the unlabeled data. We show the best results of KPT in the zero-shot setting.

**Null Prompt**   IV et al. (2022) insert a token at the end of the text (i.e. using the prompt template " $[x]$[MASK]" ) and then use the prediction of the [MASK] token to perform zero-shot classification.

**Multi-Null prompting**   Wang et al. (2021) find that simply introducing a few prompt [MASK]s can improve the performance and robustness of the Null Prompt in the zero-shot settings.

| Method | AG News | DBPedia | IMDB | Amazon | Avg. | Human/KB | Unlabeled |
|---|---|---|---|---|---|---|---|
| ManualVerb | $79.6_{0.6}$ | $71.7_{1.1}$ | $92.0_{0.7}$ | $87.3_{0.4}$ | 82.7 | ✔ | ✗ |
| Semantic Retrieval | $73.1_{1.2}$ | $78.6_{0.8}$ | $64.8_{1.3}$ | $59.4_{0.7}$ | 69.0 | ✔ | ✗ |
| NSP-BERT | $77.4_{0.6}$ | $64.7_{5.3}$ | $72.8_{1.1}$ | $72.7_{3.9}$ | 71.9 | ✔ | ✗ |
| GPT-3 w. descriptions | 83.4 | 82.5 | 88.8 | 89.4 | 86.0 | ✔ | ✗ |
| LOTClass w/o. self train | 82.2 | 86.0 | 80.2 | 85.3 | 83.4 | ✗ | ✔ |
| LOTClass | 86.4 | **91.1** | 86.5 | 91.6 | 88.9 | ✗ | ✔ |
| KPT | **86.7** | 87.4 | **94.0** | **94.6** | 90.7 | ✔ | ✔ |
| Null Prompt | $67.9_{2.0}$ | $56.8_{3.9}$ | $82.5_{1.5}$ | $89.4_{1.0}$ | 74.2 | ✗ | ✗ |
| Multi-Null Prompt | $68.2_{1.8}$ | $67.6_{1.8}$ | $86.6_{0.6}$ | $86.2_{2.7}$ | 77.2 | ✗ | ✗ |
| NPPrompt | $\mathbf{85.2}_{0.5}$ | $\mathbf{86.8}_{0.1}$ | $\mathbf{94.2}_{0.2}$ | $\mathbf{93.9}_{0.0}$ | **90.0** | ✗ | ✗ |

Table 4: Classification performance on four datasets with average results and standard error. Human: with human efforts to write deceptions or design label words. KB: with external knowledge base; Unlabeled: with unlabeled corpus. Notice that our method achieves the best performance in a fully zero-shot setting, with an absolute improvement of 12.8%. Surprisingly, it even approaches the best result with human effort/knowledge base and extra raw data.

| | MNLI (acc) | MNLI-mm (acc) | SST-2 (acc) | QNLI (acc) | RTE (acc) | MRPC (F1) | QQP (F1) | CoLA (Matt.) | Avg. |
|---|---|---|---|---|---|---|---|---|---|
| *With human designed prompts / few-shot data* | | | | | | | | | |
| Manual Label | 50.8 | 51.7 | 83.6 | 50.8 | 51.3 | 61.9 | 49.7 | 2.0 | 50.2 |
| In-context learning | $\mathbf{52.0}_{0.7}$ | $\mathbf{53.4}_{0.6}$ | $84.8_{1.3}$ | $53.8_{0.4}$ | $60.4_{1.4}$ | $45.7_{6.0}$ | $36.1_{5.2}$ | $-1.5_{2.4}$ | 48.1 |
| Auto-L | $41.6_{5.4}$ | $42.3_{6.2}$ | $84.3_{3.3}$ | $57.9_{3.9}$ | $\mathbf{61.9}_{7.5}$ | $\mathbf{67.7}_{7.9}$ | $55.5_{5.0}$ | $1.2_{4.8}$ | 51.6 |
| AMuLaP | $50.8_{2.1}$ | $52.3_{1.8}$ | $\mathbf{86.9}_{1.6}$ | $53.1_{2.8}$ | $58.9_{7.9}$ | $56.3_{5.0}$ | $60.2_{2.7}$ | $2.3_{1.4}$ | 52.6 |
| Few-shot fine-tuning | $45.8_{6.4}$ | $47.8_{6.8}$ | $81.4_{3.8}$ | $\mathbf{60.2}_{6.5}$ | $54.4_{3.9}$ | $76.6_{2.5}$ | $\mathbf{60.7}_{4.3}$ | $\mathbf{33.9}_{14.3}$ | **57.6** |
| *Fully zero-shot* | | | | | | | | | |
| Majority | 32.7 | 33.0 | 50.9 | 49.5 | 52.7 | **81.2** | 0.0 | 0.0 | 37.5 |
| Null Prompt | $33.1_{0.4}$ | $33.8_{0.5}$ | $79.1_{4.0}$ | $50.7_{0.1}$ | $47.2_{0.6}$ | $12.9_{7.0}$ | $1.3_{1.0}$ | $-1.1_{2.0}$ | 32.1 |
| Multi-Null Prompt | $38.0_{3.5}$ | $38.5_{4.1}$ | $70.2_{7.7}$ | $52.2_{1.7}$ | $53.0_{2.2}$ | $19.9_{8.7}$ | $25.5_{13.4}$ | $\mathbf{6.2}_{2.0}$ | 37.9 |
| NPPrompt | $\mathbf{45.7}_{0.6}$ | $\mathbf{45.9}_{0.5}$ | $\mathbf{86.3}_{1.2}$ | $\mathbf{57.6}_{0.7}$ | $\mathbf{55.0}_{3.4}$ | $79.8_{1.6}$ | $\mathbf{52.4}_{0.4}$ | $4.9_{4.1}$ | **53.5** |

Table 5: The performance of NPPrompt with RoBERTa-large on GLUE benchmark against other methods, including few-shot learning methods. Manual Label: using the human-designed prompts in Gao et al. (2021); In-context learning: using the in-context learning proposed in Brown et al. (2020) with RoBERTa-large; Auto-L: method in Gao et al. (2021); AMuLaP: method in Wang et al. (2022); Majority: majority class.

## 5.3   MAIN RESULTS

We demonstrate our experimental results in Table 4. Overall NPPrompt outperforms Null Prompt and Multi-Null Prompt remarkably by over 10 percent in a fully zero-shot setting. NPPrompt achieves an accuracy of over 85% on AG News and DBPedia and over 90% on IMDB and Amazon. We conjecture that topic classifications in AG News and DBPedia are more complicated than binary sentiment classifications in IMDB and Amazon, hence the higher accuracy on the latter.

NPPrompt is only slightly worse than KPT but outperforms most baseline methods in which human efforts/external knowledge or unlabeled data are strictly required. It's worth noting that NPPrompt performs much better than ManualVerb, suggesting that the label words generated by our method are more comprehensive and unbiased than human-designed ones. Besides, NPPrompt can beat GPT-3 by 4% in terms of average accuracy, a strong sign of the great potential for RoBERTa-large with 355M parameters compared to 175B parameters giant GPT-3.

To explore how our method NPPrompt performs on different kinds of tasks, we also conduct experiments on GLUE benchmark (Wang et al., 2018). Specifically, we test on Multi-Genre Natural Language Inference Matched (MNLI), Multi-Genre Natural Language Inference Mismatched (MNLI-mm)(Williams et al., 2018) , Question Natural Language Inference (QNLI) (Rajpurkar et al., 2016) and Recognizing Textual Entailment (RTE) (Bentivogli et al., 2009) for Natural Language Inference (NLI); Microsoft Research Paraphrase Matching (MRPC) (Dolan & Brockett, 2005) and Quora Question Pairs (QQP) (Chen et al., 2018) for Paraphrase Similarity Matching; Stanford Sentiment Treebank (SST-2) (Socher et al., 2013) for Sentiment Classification; The Corpus of Linguistic Acceptability (CoLA) (Warstadt et al., 2019) for Linguistic Acceptability.

As shown in Table 5, NPPrompt outperforms all other methods in fully zero-shot setting. Auto-L (Gao et al., 2021) and AMuLaP (Wang et al., 2022) are both automatic label words searching methods utilizing few-shot examples. Our method NPPrompt can even outperform them without any unlabeled data or few-shot training examples.

### 5.4 EFFECTS OF SIMILARITY FUNCTIONS IN NONPARAMETRIC AGGREGATION

Both weight and similarity functions play a critical role in the design of NPPrompt and we test how NPPrompt performs on AG News with different configurations. The "Default" setting is as stated in Equation 3 and 5. We fix the similarity function $S\left(\mathbf{emb}(v_i), \mathbf{emb}(y_j)\right) = \frac{\mathbf{emb}(v_i)}{\|\mathbf{emb}(v_i)\|} \cdot \frac{\mathbf{emb}(y_j)}{\|\mathbf{emb}(y_j)\|}$, set $w(v_i, y_j) = 1$ for the "Same weight" setting and $w(v_i, y_j) = \frac{S(\mathbf{emb}(v_i), \mathbf{emb}(y_j))}{\sum_{v_k \in \mathcal{M}(y_j)} S(\mathbf{emb}(v_k), \mathbf{emb}(y_j))}$ for the "Average weight" setting. Besides cosine similarity, the Euclidean distance and the dot product are also common similarity measures for embeddings. Consequently, we fix the weight $w(v_i, y_j) = 1$, choose

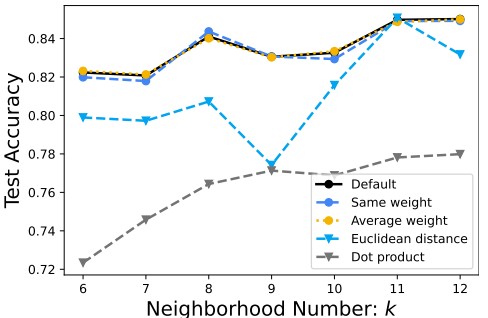

Figure 2: Effects of different aggregation.

$S\left(\mathbf{emb}(v_i), \mathbf{emb}(y_j)\right) = -\|\mathbf{emb}(v_i) - \mathbf{emb}(y_j)\|$ for the "Euclidean distance" setting and $S\left(\mathbf{emb}(v_i), \mathbf{emb}(y_j)\right) = \mathbf{emb}(v_i) \cdot \mathbf{emb}(y_j)$ for the "Dot product" setting. It can be informed from Figure 2 that with a fixed similarity function, different weight calculations yield comparable results, but with a fixed weight, cosine similarity is the optimal similarity measure.

### 5.5 CAN WE SUM OVER PROBABILITIES?

NPPrompt sums up all logits for a label word set as shown in Equation 4. Another possible approach is to sum up the probabilities from PLM's prediction for the label words and choose the argmax for all different labels as the prediction: $P(y_j|x_{\text{prompt}}) = \sum_{v_i \in \mathcal{M}(y_j)} w(v_i, y_j) \cdot P(\text{[MASK]} = v_i|x_{\text{prompt}})$, $\widetilde{y} = \arg\max_{y_j} P\left(y_j \mid x_{\text{prompt}}\right)$ We conduct experiments on AG News to compare the above two approaches, one that sums up logits ("sum logit") and one that sums up probabilities ("sum prob"). Figure 3 presents the results and we find that "sum logit" performs better at small $k$ but "sum prob" delivers

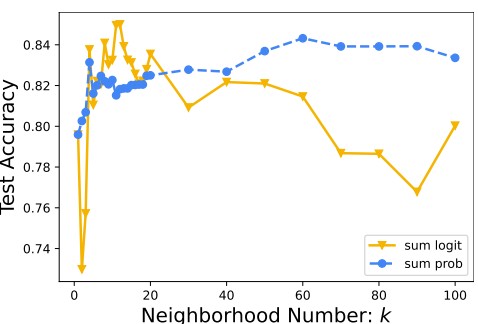

Figure 3: Test results on AG News.

better results when $k$ exceeds 30. "sum logit" achieves the best result at $k = 12$ among all experiments.

## 5.6 HOW MANY LABEL WORDS SHOULD WE CHOOSE?

Figure 4: Test results of NPPrompt for four PLMs with different neighborhood numbers.

The number of the label words impacts the performance of our method NPPrompt as well. In Figure 4, we display the performances of different models with varied neighborhood numbers. In general, NPPrompt attains similar test accuracy across different neighborhood numbers. Regardless of the choice for neighborhood number, NPPrompt-RoBERTa-large achieves over 80% accuracy in topic classification tasks on AG News and DBPedia, and it gains over 90% accuracy in sentiment classification tasks on IMDB and Amazon. In real-world applications, we can simply choose a fixed neighborhood number (e.g. 8-10) to achieve decent performance.

## 5.7 HOW DOES NPPROMPT PERFORM WITH DIFFERENT PLMS?

| Method | AG News | | DBPedia | | IMDB | | Amazon | | Avg. |
|---|---|---|---|---|---|---|---|---|---|
| NPPrompt-BERT-base | 79.4 | 30 | 77.8 | 30 | 57.7 | 4 | 53.5 | 180 | 67.1 |
| NPPrompt-BERT-large | 82.7 | 8 | 80.9 | 30 | 81.6 | 10 | 80.8 | 210 | 81.5 |
| NPPrompt-RoBERTa-base | 75.3 | 11 | 82.8 | 17 | 88.7 | 7 | 83.9 | 250 | 82.7 |
| NPPrompt-RoBERTa-large | **85.0** | 12 | **86.8** | 7 | **94.1** | 500 | **93.9** | 170 | **90.0** |

Table 6: The zero-shot results of different backbones. We also include the best neighborhood number $k$ as the second column in each category. NPPrompt-RoBERTa-large performs the best in all datasets.

NPPrompt highly depends on the pre-trained language model. The label words for the categories with various PLMs are different, a result of their unique initial word embedding and vocabularies. To study the effect of applying different PLMs, we conduct extra experiments using BERT-base-cased, BERT-large-cased, and RoBERTa-base models. The results are shown in Table 6. NPPrompt with RoBERTa-large generates the best performance, which may result from the fact that RoBERTa-large has the largest number of parameters and that it is pre-trained on the largest corpus. In general, larger models (RoBERTa-large/BERT-large) achieve better performances than base models (RoBERTa-base/BERT-base) as expected, and RoBERTa shows better accuracy than BERT models on average.

## 5.8 WHAT LABEL WORDS DO DIFFERENT PLMS CHOOSE?

We summarize the label words of different PLMs for SCHOOL category in DBPedia in Table 7. RoBERTa-large and RoBERTa-base share similar sets of label words yet with a minor discrepancy

| RoBERTa-large | | RoBERTa-base | | BERT-large | | BERT-base | |
|---|---|---|---|---|---|---|---|
| Word | Sim | Word | Sim | Word | Sim | Word | Sim |
| " school" | 1.00 | " school" | 1.00 | "school" | 1.00 | "school" | 1.00 |
| " School" | 0.80 | " School" | 0.75 | "School" | 0.69 | "School" | 0.70 |
| " schools" | 0.77 | " schools" | 0.71 | "schools" | 0.63 | "schools" | 0.63 |
| "school" | 0.74 | "school" | 0.70 | "college" | 0.55 | "college" | 0.54 |
| " SCHOOL" | 0.69 | "School" | 0.70 | "university" | 0.50 | "university" | 0.51 |
| "School" | 0.68 | " SCHOOL" | 0.56 | "student" | 0.42 | "College" | 0.40 |
| " university" | 0.66 | " college" | 0.50 | "church" | 0.41 | "church" | 0.40 |
| " college" | 0.65 | " university" | 0.50 | "house" | 0.38 | "student" | 0.37 |
| " Schools" | 0.65 | " Schools" | 0.49 | "education" | 0.38 | "students" | 0.37 |
| " schooling" | 0.64 | " schooling" | 0.45 | "students" | 0.37 | "Schools" | 0.37 |
| " preschool" | 0.63 | " preschool" | 0.44 | "class" | 0.37 | "academy" | 0.37 |
| " kindergarten" | 0.63 | " kindergarten" | 0.41 | "town" | 0.37 | "class" | 0.36 |
| " classroom" | 0.60 | " student" | 0.41 | "College" | 0.36 | "education" | 0.36 |
| " student" | 0.58 | " students" | 0.39 | "Schools" | 0.36 | "University" | 0.35 |
| " education" | 0.58 | " classroom" | 0.38 | "work" | 0.35 | "house" | 0.35 |

Table 7: The top 15 similar words of SCHOOL category in the DBPedia dataset. Sim: similarity scores.

between their similarity scores. RoBERTa-large usually produces larger similarities than RoBERTa-base. In contrast, the label words in RoBERTa are quite different from those in BERT.

## 6 DISCUSSION

NPPrompt achieves superior results in zero-shot text classifications. We attribute good performance to two aspects. Firstly, compared to fixed words or human-designed label words, using the initial word embedding from PLMs enables us to find cognates of the label words. For example, we have {" school", " School", " schools", " SCHOOL"...} for the SCHOOL category, as shown in Table 7. Secondly, we effectively elicit the potential of pre-trained language models. During the pre-trained process, language models are required to predict the masked token. The prediction of the [MASK] token of the PLM is not fixed in the inference stage, so that there is no standard correct answer to fit into the context and instead, multiple words sharing similar meanings can be predicted. Our approach reformulates the zero-shot classification problem to the masked token prediction problem which is well aligned with the pre-training process.

NPPrompt points out a promising way to deal with dynamic and open zero-shot classification problems where new classes can emerge or old classes should be deleted. Efficient PLMs and category names are all we need. Together with the key words design in Equation 7, NPPrompt can also work in special scenarios where label names do not have semantic meaning (e.g. category with label name {"A", "B", "C"}). This technique can be widely deployed in real-world applications.

**Limitations and Future Directions** For those label names without semantic meanings, several keywords are still required for NPPrompt to work well. In addition, we only focus on the zero-shot setting. There are more to explore when considering the few-shot examples in many practical applications. Besides, we only test on text classification and NLU tasks from the GLUE benchmark. Whether NPPrompt works on other tasks such as ranking or relation extraction remains an open question.

## 7 CONCLUSION

In this paper, we propose NPPrompt, a novel and effective method for fully zero-shot learning with pre-trained language models. We use initial word embedding of PLM to automatically find related words for category names, which enables us to construct the verbalizers without manual design or unlabeled corpus. Experimental results show that NPPrompt outperforms the previous zero-shot methods by large margins.

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

## A APPENDIX

### A.1 EXPERIMENTAL DETAILS

Table 8 shows all the manual templates of NSP-BERT. Table 9 summarizes manual designed descriptions of each dataset for Semantic Retrieval. As for GPT-3, we query the OpenAI API[1] and test with Davinci model. The prompts for GPT-3 are shown in Table 10. We list all templates and label names for NPPrompt of all experiments in Table 11. We also list the related words result in sentiment classification (GOOD/BAD) and NLI (YES/NO)) tasks in Table A.1.

| Dataset | Template |
|---------|----------|
| AG News | News: *label name*. |
| DBPedia | News: *label name*. |
| IMDB | This text shows *label name* sentiment. |
| Amazon | The attitude of this text is *label name*. |

Table 8: The templates of NSP-BERT (Sun et al., 2021) in Table 4.

---

[1] https://openai.com/api/

| Descriptions |
| --- |
| *AG News*: |
| The politics category is related to politics, government, and law. |
| The sports category is related to sports, competition, and athletics. |
| The business category is related to business, portfolio, economics, and money. |
| The technology category is related to technology, software, system, and science. |
| *DBPedia*: |
| The company category is related to company, corporation, enterprise, brand, and business. |
| The school category is related to school, academy, university, and college. |
| The artist category is related to artist, art, painter, musician, singer, and creative. |
| The athlete category is related to athletes, sports, Olympic, and gym. |
| The politics category is related to politics, government, and law. |
| The transportation category is related to transportation, transport, vehicle, and traffic. |
| The building category is related to buildings, construction, and structure. |
| The mountain category is related to river, lake, bay, and mountain. |
| The village category is related to village, town, and rural. |
| The animal category is related to animal, wildlife, and nature. |
| The plant category is related to plant, shrub, tree, and forest. |
| The album category is related to album, lyrics, cd, and song. |
| The film category is related to film, movie, cinema, and video. |
| The book category is related to book, novel, and publication. |
| *IMDB*: |
| The bad category is related to negative and bad reviews. |
| The good category is related to positive and good reviews. |
| *Amazon*: |
| The bad category is related to negative and bad reviews. |
| The good category is related to positive and good reviews. |

Table 9: Descriptions for Semantic Retrieval in Table 4.

| Prompts for GPT-3 |
| --- |
| *AG News* : |
| [Descriptions] Definition: In this task, you are given a sentence. Your job is to classify the following sentence into one of the four different categories. The categories are: "politics", "sports", "business", and "technology". Input: [x]. Output: |
| *DBPedia*: |
| [Descriptions] Definition: In this task, you are given a sentence. Your job is to classify the following sentence into one of the fourteen different categories. The categories are: "company", "school", "artist", "athlete", "politics", "transportation", "building", "mountain", "village", "animal", "plant", "album", "film", and "book". Input: [x]. Output: |
| *IMDB*: |
| [Descriptions] Definition: In this task, you are given a sentence. Your job is to classify the following sentence into one of the two categories. The categories are: "bad" and "good". Input: [x]. Output: |
| *Amazon*: |
| [Descriptions] Definition: In this task, you are given a sentence. Your job is to classify the following sentence into one of the two categories. The categories are: "bad" and "good". Input: [x]. Output: |

Table 10: Prompts for GPT-3 with descriptions [Descriptions] from Table 9 and input text [x].

| Dataset | Template | Label Names | $k$ |
|---|---|---|---|
| AG News | A [MASK] news : $x$ . | category 1: *world, politics*
category 2: *sports*
category 3: *business*
category 4: *technology, science* | 12 |
| DBPedia | $x_1$ $x_2$ In this sentence, $x_1$ is a [MASK] . | category 1: *company*
category 2: *school*
category 3: *artist*
category 4: *sports*
category 5: *politics, office*
category 6: *transportation, car, bus, train*
category 7: *building, construct, room, tower*
category 8: *river, lake, mountain*
category 9: *village*
category 10: *animal, pet*
category 11: *plant*
category 12: *album*
category 13: *film*
category 14: *book, publication* | 7 |
| IMDB | $x$ All in all, it was [MASK] . | positive: *good*
negative: *bad* | 500 |
| Amazon | $x$ All in all, it was [MASK] . | positive: *good*
negative: *bad* | 170 |
| SST-2 | $x_1$ It was [MASK] . | positive: *great*
negative: *terrible* | 9 |
| MNLI | $x_1$ ? [MASK] , $x_2$ | entailment: *yes*
neutral: *maybe*
contradiction: *no* | 4 |
| MNLI-mm | $x_1$ ? [MASK] , $x_2$ | entailment: *yes*
neutral: *maybe*
contradiction: *no* | 4 |
| QNLI | $x_1$ ? [MASK] , $x_2$ | entailment: *Yes, Indeed, Overall*
not_entailment: *No, Well, However* | 3 |
| RTE | $x_1$ ? [MASK] , $x_2$ | entailment: *Yes*
not_entailment: *No* | 10 |
| MRPC | $x_1$ [MASK] , $x_2$ | equivalent: *Yes*
not_equivalent: *No* | 9 |
| QQP | $x_1$ [MASK] , $x_2$ | equivalent: *Yes*
not_equivalent: *No* | 9 |
| CoLA | $x_1$ This is [MASK] . | grammatical: *true*
not_grammatical: *wrong* | 7 |

Table 11: Templates and label names for NPPrompt. $k$ refers to the best neighborhood number for RoBERTa-large.

| GOOD | | BAD | | YES | | NO | |
|---|---|---|---|---|---|---|---|
| Word | Sim | Word | Sim | Word | Sim | Word | Sim |
| " good" | 1.00 | " bad" | 1.00 | " Yes" | 1.00 | " No" | 1.00 |
| " Good" | 0.73 | " Bad" | 0.71 | " yes" | 0.79 | " no" | 0.80 |
| " GOOD" | 0.72 | " terrible" | 0.69 | " YES" | 0.73 | "No" | 0.74 |
| "good" | 0.69 | " BAD" | 0.69 | "Yes" | 0.72 | " NO" | 0.70 |
| " great" | 0.66 | " horrible" | 0.68 | " Yeah" | 0.72 | " Nope" | 0.62 |
| " excellent" | 0.66 | "bad" | 0.65 | " Yep" | 0.65 | " Yes" | 0.62 |
| " decent" | 0.66 | " awful" | 0.64 | " Sure" | 0.62 | "no" | 0.61 |
| "Good" | 0.65 | "Bad" | 0.64 | " No" | 0.62 | " Nobody" | 0.59 |
| " nice" | 0.64 | " good" | 0.63 | " Indeed" | 0.61 | " Nos" | 0.57 |
| " bad" | 0.63 | " badly" | 0.62 | " yeah" | 0.60 | " The" | 0.57 |
| " better" | 0.62 | " crappy" | 0.60 | "yes" | 0.59 | " Yeah" | 0.57 |
| " wonderful" | 0.58 | " lousy" | 0.60 | " Wow" | 0.59 | " Nothing" | 0.56 |
| " best" | 0.58 | " worst" | 0.60 | " Absolutely" | 0.58 | " Not" | 0.56 |
| " terrific" | 0.57 | " horrendous" | 0.60 | " Nope" | 0.58 | " Never" | 0.56 |
| " fantastic" | 0.57 | " worse" | 0.59 | " Okay" | 0.57 | " None" | 0.55 |
| " mediocre" | 0.57 | " nasty" | 0.59 | " Oh" | 0.57 | " Number" | 0.55 |
| " lousy" | 0.57 | " shitty" | 0.59 | " Hello" | 0.57 | " So" | 0.54 |
| " satisfactory" | 0.56 | " dreadful" | 0.59 | " Hey" | 0.57 | " Any" | 0.54 |
| " marvelous" | 0.56 | " rotten" | 0.58 | " Nevertheless" | 0.57 | " And" | 0.54 |
| " GREAT" | 0.56 | " harmful" | 0.58 | " However" | 0.56 | "NO" | 0.53 |

Table 12: The top 20 similar words of label names in sentiment classification (GOOD/BAD) and NLI (YES/NO) tasks.

## A.2 EXPERIMENTS OF NPPROMPT WITH T5 MODEL

NPPrompt also works on text-to-text pre-trained language models (e.g. T5 (Raffel et al., 2020)) with minor modification. We use T5-base to generate the missing spans at the end of the prompt text. We choose the first predicted token as the input to the verbalizer and follow the nonparametric aggregation steps to decide the category. The results are shown in Table A.1. NPPrompt-T5-base shows good performance but it can not outperform NPPrompt-RoBERTa-large.

| Dataset | Template | $k$ | Accuracy |
|---|---|---|---|
| AG News | $x$ In this sentence, the topic is about [MASK] | 15 | $76.8_{2.3}$ |
| DBPedia | $x_1$ $x_2$ In this sentence, $x_1$ is a [MASK] | 15 | $78.3_{1.8}$ |
| IMDB | $x$ In summary, the movie was [MASK] | 15 | $68.5_{2.1}$ |
| Amazon | $x$ All in all, it was [MASK] | 15 | $65.3_{3.2}$ |

Table 13: Prompt template and test results of NPPrompt with T5-base.

## A.3 EXPERIMENTS OF NPPROMPT ON MULTIPLE CHOICE QUESTION ANSWERING TASK

We test NPPrompt on the CommonsenseQA (CQA) dataset (Talmor et al., 2019), a widely used multiple-choice question answering (QA) task. In this new setting, we use the prompt template "$x$ The answer is [MASK].", e.g. "What do animals do when an enemy is approaching? The answer is [MASK].". Then we search for $k$-nearest neighbors for each target answer with $k = 15$. Finally, we follow the process when we deal with text classification tasks and obtain the prediction. The experiment results are listed in Table A.1 (few-shot results from Zelikman et al. (2022)). NPPrompt not only achieves satisfactory results on CommonsenseQA dataset but even outperforms few-shot GPT-J (Wang, 2021) as well.

| Method | CQA Dev Set Accuracy |
|---|---|
| Few-shot Direct GPT-J | 20.9 |
| Few-shot CoT GPT-J | 36.6 |
| Few-shot CoT LaMDA 137B | 55.6 |
| NPPrompt-RoBERTa-large | 34.2 |

Table 14: Test results on CommonsenseQA dataset. Direct: directly output the final answer; CoT: prompted with chain-of-thought (CoT) rationales; LaMDA: method in Wei et al. (2022).

## A.4 EXTENSION TO MULTI-WORD EXPRESSIONS

Here we extend our method to support multi-word label names like NATURALPLACE, MEANOF-TRANSPORTATION and etc. The major part is to obtain related words to a multi-word label name. Once we obtain the related words, the rest non-parametric aggregation step remains identical. We consider two scenarios:

**The label name is multi-word (i.e., phrase) and related words are still single-words**   To model the phrase, we use average contextualized embedding instead of word embedding for both label names and related single-words to compute cosine similarity. As suggested in Su et al. (2021), we whiten the contextualized output of RoBERTa by a linear transformation obtained from the contextualized embedding of all words in vocabulary. To obtain the best result, we select the output of layer 6 of RoBERTa. This extension achieves 61% accuracy on the DBpedia dataset using the original multi-word label names (original label names can be found at `https://rdrr.io/cran/textdata/man/dataset_dbpedia.html`).

**Both the label name and related words are phrases**   Since the search space of a related phrase is exponentially large in its length, we use another prompt to filter candidate words. The template we use is "[LABEL_NAME] can also be called [MASK]$*n$.", where $n$ is the length of the candidate. For example, if the label name is MEANOFTRANSPORTATION and $n = 2$, the template will look like "Mean of transportation can also be called [MASK] [MASK].". We feed it to RoBERTa and filter top-$k$ candidate phrases of masked prediction. Since masked prediction is conditionally independent of each mask, we further re-rank the top-$k$ candidate phrases by either the contextualized embedding method mentioned above or another auto-regressive LM. For the latter one, we evaluate the perplexity of the template with [MASK] filled by candidate phrases. This generates 71% accuracy on DBpedia if the length of the phrase is two and the re-ranking is performed by GPT-2 (Radford et al., 2019).

