# OpenReview forum: "Pre-trained Language Models can be Fully Zero-Shot Learners"
_ICLR.cc/2023/Conference — Submitted to ICLR 2023_

### Official Review · Reviewer_Gywf · 2022-10-24

**Confidence:** 3
**Correctness:** 4
**Technical Novelty And Significance:** 3
**Empirical Novelty And Significance:** 4
**Recommendation:** 6

**Clarity, Quality, Novelty And Reproducibility:**

The paper was well written and easy to follow. I would require authors to add the github link for the code.

**Strength And Weaknesses:**

The authors put significant effort on proving effectiveness of their method in a variety of NLP tasks. However, I wanted to see significant test results to make sure that the improvements are not random.

**Summary Of The Paper:**

The authors propose a new language model named non parametric prompting PLM for natural language understanding specially for zero-shot learning. It is an important topic because these days many word-class associations are being produced by end users and previous models heavily depend on unlabeled data and human effort.  The authors showed that the proposed method outperforms state-of-the-art in terms of text classification accuracy and GLUE benchmarks on four different datasets including AG news, DBPedia, IMDB. and Amazon.

**Summary Of The Review:**

Overall, zero shot learning is an interesting topic in natural language processing as so many new categories and topics are being produced on the web. The authors proposed a simple and easy to implement method for pre trained language models to minimize human effort in terms of labeling and building training data. Overall I am satisfied with the current draft of the paper and request to move forward with discussion.

---

> ### Author Response · Authors · 2022-11-10
> **Reply to Reviewer Gywf Comments**
>
> Thank you for your positive feedback! We upload a paper revision and we color the revised part in blue. We provide the following responses to your concerns.
>
> **Q1. However, I wanted to see significant test results to make sure that the improvements are not random.**
>
> A1. Per your suggestion, we use ten bootstrap test datasets to show the randomness of our method. We report the average accuracy and standard error in the revised paper. We can not get significant test results between our method and baselines (Null Prompt and Multi-Null Prompt) since baselines didn’t release their code. However, based on the mean and variance, our method NPPrompt is robust to bootstrap resampling and outperforms the previous best fully zero-shot method by big margins, with absolute gains of 12.8% in accuracy on text classification and 15.6% on the GLUE benchmark.
>
> **Q2. I would require authors to add the github link for the code.**
>
> Q2. We already upload our code to [https://anonymous.4open.science/r/NPPrompt](https://anonymous.4open.science/r/NPPrompt)

---

### Official Review · Reviewer_LMgP · 2022-10-25

**Confidence:** 3
**Correctness:** 3
**Technical Novelty And Significance:** 2
**Empirical Novelty And Significance:** 2
**Recommendation:** 6

**Clarity, Quality, Novelty And Reproducibility:**

This paper is well-written. The work is original, but the proposed method is kind of similar to many existing works, which makes it less novel.

**Strength And Weaknesses:**

Strength
1. Proposed method is simple, intuitive and effective.
2. Conduct extensive experiments on a wide range of NLU tasks.

Weaknesses
1. The proposed method is not very novel. Generalizing LMs to NLU tasks by checking the vocabulary distribution over the masked token is a common practice. The method which leverages related words to label category does not look very novel to me.

**Summary Of The Paper:**

This paper studies how to effectively transfer pretrained language models to natural language understanding (NLU) tasks in a zero-shot manner. The proposed method, NPPrompt, does not require any labeled sample or rely on humans to construct prompt label words. NPPrompt generates the label words by searching the related words from the initial word embedding of the pre-trained language model. Then, it aggregates logits from the label words and predicts the category with the largest score. Experiments show that the proposed method is effective and outperforms strong baselines with a large margin.

**Summary Of The Review:**

This paper proposed NPPrompt, which aims to enable LMs' zero-shot ability to NLU tasks without requiring any labeled sample or relying on humans to construct prompt label words. Results show that NNPrompt outperforms strong baselines with a large margin. However, the overall novelty of this paper is limited.

---

> ### Author Response · Authors · 2022-11-10
> **Reply to Reviewer LMgP Comments**
>
> Thank you for your positive feedback! Below, we present our response to the questions:
>
> **Q1. “The proposed method is not very novel.” “The work is original, but the proposed method is kind of similar to many existing works, which makes it less novel.” “The overall novelty of this paper is limited.”**
>
> A1. We study the fully zero-shot problem where only the target label names are available. This is the minimal requirement to define a classification task. Existing works on fully zero-shot settings utilize multiple null prompts to search the label words (Wang et al, 2021). However, we take a different approach which effectively elicits information from pre-trained language models using masked prompt prediction and nonparametric aggregation. Specifically, we search over the whole vocabulary $V$ for the top-$k$ words nearest to the label name in the PLM’s embedding space. Then we take nonparametric aggregation of the predicted logits on the [MASK] token to obtain the probability on any labels.
>
> The empirical results show that our method outperforms the previous best fully zero-shot method by big margins and has a comparable performance with the zero-shot methods which need additional information.
>
> In the real world, there are dynamic and open zero-shot classification problems where new classes can emerge or old classes should be deleted. For instance, if Twitter wants to find trending topics for this week they need a zero-shot classifier with minimum human effort and labeled/unlabeled data. NPPrompt uses only pre-trained language models and does not require any labeled data or additional raw corpus for further fine-tuning, nor does it rely on humans to construct a comprehensive set of prompt label words.

---

### Official Review · Reviewer_DKg9 · 2022-11-04

**Confidence:** 4
**Correctness:** 2
**Technical Novelty And Significance:** 3
**Empirical Novelty And Significance:** 2
**Recommendation:** 5

**Clarity, Quality, Novelty And Reproducibility:**

- The paper is mostly clear and the novelty is incremental. I think the reproducibility should be okay since the method is pretty naive.

- Can you show the kNN results for sentiment analysis tasks and NLI tasks? I don't find them in the paper but they are quite important for qualitative analysis.

**Strength And Weaknesses:**

## Strengths

- The proposed NPPrompt method is simple and effective. It only requires the label strings for doing zero-shot classification.
- The empirical results show that it is on the same level as other methods that need more information about the labels.

## Weakness

- The proposed method is pretty limited to the classification tasks, particularly for news classification, where the label names are informative and have many semantic relevant words in the vocabulary. It is also limited when the label names are multi-word expressions or phrases. It does not support the cases where the label names are not informative enough and additional label description is needed.

- The title and narrative can be misleading and an overclaim. As mentioned before, it is limited to masked LMs and the advantage is mainly for tasks like news classification.

Suggestion:

- I suggest the authors extend the proposed method to *multi-label classification* settings where an example can have multiple acceptable news.
- A major limitation is the multi-word expression and out-of-vocabluary label names.  How do you generalize the method to support that?
- Also, I suggest the authors try some multiple-choice QA problems.

**Summary Of The Paper:**

The paper presents a novel method for zero-shot inference with masked language models (MLMs) such as BERT and RoBERTa for classification problems. The key idea is to use the encoders of the MLMs themselves to find relevant words for a given label name. For example, if the target label is "SPORTS", then the proposed method will first find the words by searching k-nearest neighbors with embedding distance. The embeddings are induced by the MLM itself. Finally, they aggregate the logits of these k words (for filling the masked positions). The authors argue that this new method (named NPPrompt) is a fully zero-shot classification method because it does not need any other information about labels. The empirical comparisons show that the proposed method has a comparable performance with other zero-shot methods (that needs some additional info such as KB or the unlabeled data).



**Summary Of The Review:**

Overall, I think the paper presents a novel zero-shot classification paper but the limitations are there and the performance is not that significant. I think if the authors can extend the paper with my suggestions in the above section, the paper will be more suitable to the ICLR community. Otherwise, it will have much less impact to the field.

---

> ### Author Response · Authors · 2022-11-10
> **Reply to Reviewer DKg9 Comments (Part 1)**
>
> Thanks a lot for your constructive comments and suggestions! We upload a paper revision and we color the revised part in blue. And here are our responses:
>
>
> **Q1. Limited to classification task, particularly for new classification**
>
> A1. Our method NPPrompt works for multiple NLU tasks, including news/topic classification (AG News, DBPedia), sentiment classification (IMDB, Amazon, SST-2), Natural Language Inference (MNLI, QNLI, RTE), Paraphrase Similarity Matching (MRPC, QQP), and Linguistic Acceptability (CoLA). Notice that in addition to Table 4, Table 5 lists results on GLUE with more tasks. Our experimental results in Table 4 and Table 5 show that NPPrompt outperforms other fully zero-shot methods (Null Prompt and Multi-Null Prompt) by a big margin (over 10 percent). Our method NPPrompt can even outperform several few-shot learning methods (Table 5).
>
> **Q2. Label names need to be informative**
>
> A2. We acknowledge this comment and we’ve already included a discussion in Section 6 of the paper (2nd and 3rd paragraph).
> Since the main setting of the problem is *fully zero-shot*, which is to transfer a pre-trained model to a new prediction problem without any extra data (neither annotated nor raw) or external knowledge base, we still need the label names to be semantically informative or we need the keywords for the categories. This setting is realistic in practice as we see from all tasks considered in this paper and other related papers (e.g. T0 [Sanh et al, 2022]).  If a label name is symbolic and abstract (e.g. “A”, “B”, “C”), together with the keywords design in Equation 7, NPPrompt can also have good performance.
>
> **Q3. Limited for multi-word expressions, out-of-vocabulary**
>
> A3. We already include one solution to categories with multi-word or OOV label names, using multiple single-word keywords (see Eq (7) and the paragraph above it). We also included a description in the paper (the paragraph before the Experiment section): *“There are certain conditions where one class has label names containing little semantic meaning or where several keywords are needed to define a label. For instance, in the DBPedia dataset, one class is related to NaturalPlace, then we can use the keywords {“river”, “lake”, “mountain”} to represent this class.”*
>
> Thanks for your suggestion, in our revised paper (Appendix A.4), we extend the NPPrompt method to label names with longer expressions. The major part is to find related words to multi-word label names. Once we obtain them, the rest of the non-parametric aggregation step remains identical.
>
> 1. If the label name is multi-word (i.e. a phrase) and related words are still single-words. To model the phrase, we use average contextualized embedding (e.g., the whitened output of BERT) instead of word embedding for both label names and candidates to compute cosine similarity. This extension achieves 61% accuracy on the DBpedia dataset using the original multi-word label names (original label names can be found at [https://rdrr.io/cran/textdata/man/dataset_dbpedia.html](https://rdrr.io/cran/textdata/man/dataset_dbpedia.html)).
> 2. If both the label name and candidates are phrases. Since the search space of related phrases is exponentially large in its length, we use another prompt to filter candidate phrases. One example template is *[LABEL NAME] can also be called [MASK]\*n.*, where n is the candidate length. We feed it to RoBERTa and filter top-k candidate phrases of masked prediction. Since masked prediction is conditionally independent of each mask, we further re-rank the top-k candidate phrases by either the contextualized embedding method mentioned in (1) or another auto-regressive LM. For the latter one, we evaluate the perplexity of the template with [MASK] filled by candidate phrases. This generates 71% accuracy on DBpedia if the length of the phrase is two and the reranking is performed by GPT-2.
>
> **Q3. The title and narrative can be misleading and an overclaim. As mentioned before, it is limited to masked LMs and the advantage is mainly for tasks like news classification.**
>
> A3. Thanks for your suggestion! We will revise our title to reflect the scope of *“NLU tasks”*. We add additional experimental results for other models (e.g. T5, in Appendix A.2), showing its effectiveness for different model architectures.
>
> **Q4. I suggest the authors extend the proposed method to multi-label classification settings where an example can have multiple acceptable news.**
>
> A4. Thanks for your suggestion! Our method NPPrompt can be easily extended to multi-label classification settings. Recall that the best class prediction of NPPrompt is selected from the maximum of all labels. If we set a threshold for $Q$, we can deal with multi-label classification tasks, meaning that if the text’s cumulative logits for one class are larger than the threshold, we think of this text as being related to that class.

---

> ### Author Response · Authors · 2022-11-10
> **Reply to Reviewer DKg9 Comments (Part 2)**
>
> **Q5. I suggest the authors try some multiple-choice QA problems.**
>
> A5. We test NPPrompt on the CommonsenseQA dataset [Talmor et al, 2019], a widely used multiple-choice QA task. In this new setting, we use the prompt template *“[TEXT] The answer is [MASK]”*, e.g. *“What do animals do when an enemy is approaching? The answer is [MASK]”*. Then we search for k-nearest neighbors for each target answer. Finally, we follow the process when we deal with text classification tasks and obtain the prediction. The experiment results are listed in Appendix A.3. NPPrompt not only achieves satisfactory results on the CommonsenseQA dataset but even outperforms few-shot GPT-J (Wang, 2021) as well.
>
> | **Method** | **CQA Dev Set Accuracy** |
> |---|:---:|
> |Few-shot Direct GPT-J &diams; | 20.9 |
> |Few-shot CoT GPT-J &diams; |36.6 |
> |Few-shot CoT LaMDA 137B &diams; | 55.6 |
> |NPPrompt-RoBERTa-large | 34.2 |
> &diams; Results from Zelikman et al. (2022)
>
> **Q6. Can you show the kNN results for sentiment analysis tasks and NLI tasks?**
>
> A6. We add the related words result in Appendix Table 12.
>
> | **Word** | **Sim** | **Word** | **Sim** | **Word** | **Sim** | **Word** | **Sim** |
> |---|:---:|---|:---:|---|:---:|---|:---:|
> | " good" | 1.0 | " bad" | 1.0 | " Yes" | 1.0 | " No" | 1.0 |
> | " Good" | 0.73 | " Bad" | 0.71 | " yes" | 0.79 | " no" | 0.8 |
> | " GOOD" | 0.72 | " terrible" | 0.69 | " YES" | 0.73 | "No" | 0.74 |
> | "good" | 0.69 | " BAD" | 0.69 | "Yes" | 0.72 | " NO" | 0.7 |
> | " great" | 0.66 | " horrible" | 0.68 | " Yeah" | 0.72 | " Nope" | 0.62 |
> | " excellent" | 0.66 | "bad" | 0.65 | " Yep" | 0.65 | " Yes" | 0.62 |
> | " decent" | 0.66 | " awful" | 0.64 | " Sure" | 0.62 | "no" | 0.61 |
> | "Good" | 0.65 | "Bad" | 0.64 | " No" | 0.62 | " Nobody" | 0.59 |
> | " nice" | 0.64 | " good" | 0.63 | " Indeed" | 0.61 | " Nos" | 0.57 |
> | " bad" | 0.63 | " badly" | 0.62 | " yeah" | 0.6 | " The" | 0.57 |
> | " better" | 0.62 | " crappy" | 0.6 | "yes" | 0.59 | " Yeah" | 0.57 |
> | " wonderful" | 0.58 | " lousy" | 0.6 | " Wow" | 0.59 | " Nothing" | 0.56 |
> | " best" | 0.58 | " worst" | 0.6 | " Absolutely" | 0.58 | " Not" | 0.56 |
> | " terrific" | 0.57 | " horrendous" | 0.6 | " Nope" | 0.58 | " Never" | 0.56 |
> | " fantastic" | 0.57 | " worse" | 0.59 | " Okay" | 0.57 | " None" | 0.55 |
> | " mediocre" | 0.57 | " nasty" | 0.59 | " Oh" | 0.57 | " Number" | 0.55 |
> | " lousy" | 0.57 | " shitty" | 0.59 | " Hello" | 0.57 | " So" | 0.54 |
> | " satisfactory" | 0.56 | " dreadful" | 0.59 | " Hey" | 0.57 | " Any" | 0.54 |
> | " marvelous" | 0.56 | " rotten" | 0.58 | " Nevertheless" | 0.57 | " And" | 0.54 |
> | " GREAT" | 0.56 | " harmful" | 0.58 | " However" | 0.56 | "NO" | 0.53 |
>
> ***
> Alon Talmor, Jonathan Herzig, Nicholas Lourie, and Jonathan Berant. Commonsenseqa: A question answering challenge targeting commonsense knowledge. In NAACL, 2019.
>
> Ben Wang. Mesh-Transformer-JAX: Model-Parallel Implementation of Transformer Language
> Model with JAX. https://github.com/kingoflolz/mesh-transformer-jax, May 2021.
>
> E. Zelikman, Yuhuai Wu, and Noah D. Goodman. Star: Bootstrapping reasoning with reasoning. ArXiv, abs/2203.14465, 2022.

---

### Author Response · Authors · 2022-11-17
**Do the new experiments and discussions address your concerns?**

Dear reviewers,

We greatly appreciate your comments and suggestions, and we believe your incorporation led to valuable improvements to the paper in terms of clarity and the thoroughness of the experiments. In response to your comments, we ran several new experiments and tried hard to improve our explanations. All changes and additions are typeset in blue.

Do these new experiments and explanations address your concerns? There is only a little time left to discuss and revise our paper. We are looking forward to your further feedback.

Sincerely,

The authors

---

### Decision · Program_Chairs · 2023-01-20

**Decision:**

Reject

**Justification For Why Not Higher Score:**

The novelty is rather limited -- extending the set of label words. The method is unlikely to be of practical importance as highlighted in the summary. Cherrypicking datasets on which the method will work leaves open the question of the limitation of the method.

**Justification For Why Not Lower Score:**

N/A

**Metareview: Summary, Strengths And Weaknesses:**

This paper proposes an approach to zero-shot text classification. The idea is to only take the label name as supervision, to extend the label set to similar words using embeddings and to compute the probability of the different labels using MLM-based pretrained models. The method is evaluated empirically and shown to outperform baselines on topic and sentiment classification datasets.

It is nice to see that pretrained LMs can perform zero-shot classification only using the label identity. The proposed method for extending the label set is interesting.

However, this method works in very limited settings -- only for classification and when the class label is unambiguous. It is already known in this setting that the label identity is enough to achieve non-trivial performance. While the method improves the performance even further, it is unlikely that the particular method will have any practical use as it is much simpler to either label a few examples or write a short task description (e.g., prompt) rather than verify that the task is one in which the label identity is sufficient.